# Can Brain Signals Reveal Inner Alignment with Human Languages? 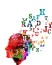

**William Han[1*], Jielin Qiu[1*], Jiacheng Zhu[1,2], Mengdi Xu[1],**
**Douglas Weber[1], Bo Li[3], Ding Zhao[1]**
[1]Carnegie Mellon University, [2]MIT CSAIL, [3]University of Chicago

## Abstract

Brain Signals, such as Electroencephalography (EEG), and human languages have been widely explored independently for many downstream tasks, however, the connection between them has not been well explored. In this study, we explore the relationship and dependency between EEG and language. To study at the representation level, we introduced **MTAM**, a **M**ultimodal **T**ransformer **A**lignment **M**odel, to observe coordinated representations between the two modalities. We used various relationship alignment-seeking techniques, such as Canonical Correlation Analysis and Wasserstein Distance, as loss functions to transfigure features. On downstream applications, sentiment analysis and relation detection, we achieved new state-of-the-art results on two datasets, ZuCo and K-EmoCon. Our method achieved an F1-score improvement of 1.7% on K-EmoCon and 9.3% on Zuco datasets for sentiment analysis, and 7.4% on ZuCo for relation detection. In addition, we provide interpretations of the performance improvement: (1) feature distribution shows the effectiveness of the alignment module for discovering and encoding the relationship between EEG and language; (2) alignment weights show the influence of different language semantics as well as EEG frequency features; (3) brain topographical maps provide an intuitive demonstration of the connectivity in the brain regions. Our code is available at https://github.com/Jason-Qiu/EEG_Language_Alignment.

## 1 Introduction

Brain activity is an important parameter in furthering our knowledge of how human language is represented and interpreted (Toneva et al., 2020; Williams and Wehbe, 2021; Reddy and Wehbe, 2021; Wehbe et al., 2020; Deniz et al., 2021). Researchers from domains such as linguistics, psychology, cognitive science, and computer science

have made large efforts in using brain-recording technologies to analyze cognitive activity during language-related tasks and observed that these technologies added value in terms of understanding language (Stemmer and Connolly, 2012).

Basic linguistic rules seem to be effortlessly understood by humans in contrast to machinery. Recent advances in natural language processing (NLP) models (Vaswani et al., 2017) have enabled computers to maintain long and contextual information through self-attention mechanisms. This attention mechanism has been maneuvered to create robust language models but at the cost of tremendous amounts of data (Devlin et al., 2019; Liu et al., 2019b; Lewis et al., 2020; Brown et al., 2020; Yang et al., 2019). Although performance has significantly improved by using modern NLP models, they are still seen to be suboptimal compared to the human brain. In this study, we explore the relationship and dependencies of EEG and language. We apply EEG, a popularized routine in cognitive research, for its accessibility and practicality, along with language to discover connectivity.

Our contributions are summarized as follows:

- To the best of our knowledge, this is the first work to explore the fundamental relationship and connectivity between EEG and language through computational multimodal methods.

- We introduced **MTAM**, a **M**ultimodal **T**ransformer **A**lignment **M**odel, that learns coordinated representations by hierarchical transformer encoders. The transformed representations showed tremendous performance improvements and state-of-the-art results in downstream applications, i.e., sentiment analysis and relation detection, on two datasets, ZuCo 1.0/2.0 and K-EmoCon.

- We carried out experiments with multiple alignment mechanisms, i.e., canonical correlation analysis and Wasserstein distance, and

---

*marked as equal contribution

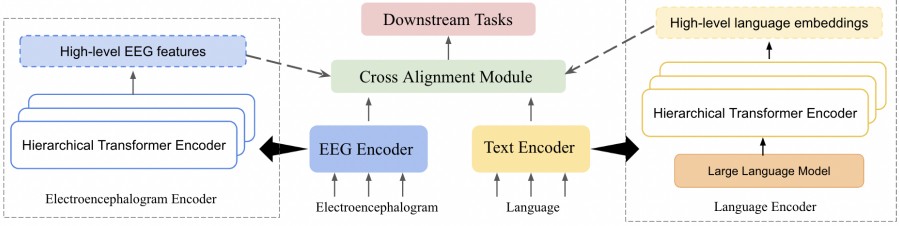

Figure 1: The architecture of our model, where EEG and language features are coordinately explored by two encoders. The EEG encoder and language encoder are shown on the left and right, respectively. The cross-alignment module is used to explore the connectivity and relationship within two domains, while the transformed features are used for downstream tasks.

proved that relation-seeking loss functions are helpful in downstream tasks.

- We provided interpretations of the performance improvement by visualizing the original & transformed feature distribution, showing the effectiveness of the alignment module for discovering and encoding the relationship between EEG and language.

- Our findings on word-level and sentence-level EEG-language alignment showed the influence of different language semantics as well as EEG frequency features, which provided additional explanations.

- The brain topographical maps delivered an intuitive demonstration of the connectivity of EEG and language response in the brain regions, which issues a physiological basis for our discovery.

## 2 Related Work

**Multimodal Learning of Language and Other Brain Signals** Wehbe et al. (2014) used a recurrent neural network to perform word alignment between MEG activity and the generated word embeddings. Toneva and Wehbe (2019) utilized word-level MEG and fMRI recordings to compare word embeddings from large language models. Schwartz et al. (2019) used MEG and fMRI data to fine-tune a BERT language model (Devlin et al., 2019) and found the relationships between these two modalities were generalized across participants. Huang et al. (2020) leveraged CT images and text from electronic health records to classify pulmonary embolism cases and observed that the multimodal model with late fusion achieved the best performance. However, the relationship between language and EEG has not been explored before.

**Multimodal Learning of EEG and Language** Foster et al. (2021) applied EEG signals to pre-

dict specific values of each dimension in a word vector through regression models. Wang and Ji (2021) used word-level EEG features to decode corresponding text tokens through an open vocabulary, sequence-to-sequence framework. Hollenstein et al. (2021) focused on a multimodal approach by utilizing a combination of EEG, eye-tracking, and text data to improve NLP tasks, but did not explore the relationship between EEG and language. More related work can be found in Appendix E.

## 3 Methods

### 3.1 Overview of Model Architecture

The architecture of our model is shown in Fig. 1. The bi-encoder architecture is helpful in projecting embeddings into vector space for methodical analysis (Liu et al., 2019a; Hollenstein et al., 2021; Choi et al., 2021). Thus in our study, we adopt the bi-encoder approach to effectively reveal hidden relations between language and EEG. The **MTAM**, Multimodal Transformer Alignment Model, contains several modules. We use a dual-encoder architecture, where each view contains hierarchical transformer encoders. The inputs of each encoder are EEG and language, respectively. For EEG hierarchical encoders, each encoder shares the same architecture as the encoder module in Vaswani et al. (2017). In the current literature, researchers assume that the brain acts as an encoder for high-dimensional semantic representations (Wang and Ji, 2021; Gauthier and Ivanova, 2018; Correia et al., 2013). Based on this assumption, the EEG signals act as low-level embeddings. By feeding it into its respective hierarchical encoder, we extract transformed EEG embeddings as input for the cross alignment module. As for the language path, the language encoder is slightly different from the EEG encoder. We first process the text with a pretrained large language model (LLM) to extract text em-

Table 1: Comparison with baselines on Zuco dataset for Sentiment Analysis (SA) and Relation Detection (SD).

| Task | Model | Sentence Level | | | | Word Level | | | | Concat Word Level | | | |
|---|---|---|---|---|---|---|---|---|---|---|---|---|---|
| | | Prec | Rec | F1 | Acc | Prec | Rec | F1 | Acc | Prec | Rec | F1 | Acc |
| Sentiment Analysis | MLP-EEG | 0.483 | 0.477 | 0.480 | 0.499 | 0.451 | 0.483 | 0.467 | 0.473 | 0.447 | 0.464 | 0.455 | 0.450 |
| | MLP-Text | 0.359 | 0.357 | 0.357 | 0.373 | 0.380 | 0.388 | 0.384 | 0.387 | 0.210 | 0.243 | 0.225 | 0.228 |
| | Bi-LSTM-EEG | 0.506 | 0.492 | 0.498 | 0.499 | 0.508 | 0.494 | 0.501 | 0.503 | 0.459 | 0.457 | 0.457 | 0.456 |
| | Bi-LSTM-Text | 0.420 | 0.347 | 0.380 | 0.371 | 0.335 | 0.326 | 0.330 | 0.329 | 0.341 | 0.322 | 0.331 | 0.329 |
| | Transformer-EEG | 0.665 | 0.659 | 0.662 | 0.662 | 0.624 | 0.630 | 0.627 | 0.664 | 0.624 | 0.630 | 0.627 | 0.614 |
| | Transformer-Text | 0.548 | 0.546 | 0.547 | 0.507 | 0.527 | 0.533 | 0.530 | 0.582 | 0.558 | 0.547 | 0.552 | 0.550 |
| | ResNet-EEG | 0.515 | 0.508 | 0.511 | 0.512 | 0.530 | 0.539 | 0.534 | 0.532 | 0.518 | 0.517 | 0.518 | 0.516 |
| | ResNet-Text | 0.214 | 0.183 | 0.165 | 0.222 | 0.198 | 0.199 | 0.198 | 0.200 | 0.202 | 0.211 | 0.206 | 0.210 |
| | RNN-Multimodal (Hollenstein et al., 2021) | — | — | — | — | 0.728 | 0.717 | 0.714 | — | — | — | — | — |
| | CNN-Multimodal (Hollenstein et al., 2021) | — | — | — | — | 0.738 | 0.724 | 0.723 | — | — | — | — | — |
| | Ours-EEG | **0.779** | **0.783** | **0.781** | **0.779** | **0.783** | **0.786** | **0.784** | **0.781** | **0.793** | **0.790** | **0.791** | **0.792** |
| | Ours-Text | **0.763** | **0.762** | **0.762** | **0.735** | **0.747** | **0.749** | **0.748** | **0.753** | **0.740** | **0.764** | **0.763** | **0.772** |
| | Ours-Multimodal | **0.822** | **0.829** | **0.826** | **0.826** | **0.821** | **0.812** | **0.816** | **0.827** | **0.802** | **0.809** | **0.806** | **0.813** |
| Relation Detection | MLP-EEG | 0.270 | 0.273 | 0.271 | 0.270 | 0.278 | 0.283 | 0.280 | 0.277 | 0.261 | 0.263 | 0.262 | 0.258 |
| | MLP-Text | 0.191 | 0.214 | 0.192 | 0.254 | 0.249 | 0.286 | 0.266 | 0.258 | 0.228 | 0.231 | 0.229 | 0.230 |
| | Bi-LSTM-EEG | 0.331 | 0.342 | 0.334 | 0.329 | 0.350 | 0.354 | 0.352 | 0.351 | 0.338 | 0.324 | 0.331 | 0.330 |
| | Bi-LSTM-Text | 0.153 | 0.173 | 0.149 | 0.186 | 0.200 | 0.199 | 0.199 | 0.201 | 0.182 | 0.133 | 0.154 | 0.148 |
| | Transformer-EEG | 0.502 | 0.440 | 0.468 | 0.479 | 0.339 | 0.341 | 0.340 | 0.358 | 0.310 | 0.316 | 0.313 | 0.315 |
| | Transformer-Text | 0.428 | 0.487 | 0.444 | 0.420 | 0.488 | 0.491 | 0.489 | 0.487 | 0.301 | 0.299 | 0.300 | 0.300 |
| | ResNet-EEG | 0.359 | 0.399 | 0.413 | 0.390 | 0.349 | 0.350 | 0.350 | 0.350 | 0.352 | 0.363 | 0.358 | 0.352 |
| | ResNet-Text | 0.314 | 0.283 | 0.265 | 0.322 | 0.311 | 0.336 | 0.323 | 0.326 | 0.278 | 0.289 | 0.283 | 0.288 |
| | CNN-Multimodal (Hollenstein et al., 2021) | — | — | — | — | 0.647 | 0.664 | 0.650 | — | — | — | — | — |
| | RNN-Multimodal (Hollenstein et al., 2021) | — | — | — | — | 0.652 | 0.690 | 0.668 | — | — | — | — | — |
| | Ours-EEG | **0.676** | **0.698** | **0.687** | **0.673** | **0.669** | **0.681** | **0.678** | **0.679** | **0.658** | **0.690** | **0.669** | **0.669** |
| | Ours-Text | **0.534** | **0.538** | **0.536** | **0.526** | **0.534** | **0.565** | **0.552** | **0.549** | **0.455** | **0.494** | **0.472** | **0.473** |
| | Ours-Multimodal | **0.742** | **0.746** | **0.742** | **0.744** | **0.741** | **0.743** | **0.742** | **0.746** | **0.737** | **0.734** | **0.736** | **0.737** |

beddings and then use hierarchical transformer encoders to transform the raw text embeddings into high-level features. The mechanism of the cross alignment module is to explore the inner relationship between EEG and language through a connectivity-based loss function. In our study, we investigate two alignment methods, i.e., Canonical Correlation Analysis (CCA) and Wasserstein Distance (WD). The output features from the cross alignment module can be used for downstream applications. The details of each part are introduced in Appendix B.3.

# 4 Experiments

## 4.1 Experimental Results and Discussions

In this study, we evaluate our method on two downstream tasks: Sentiment Analysis (SA) and Relation Detection (RD) of two datasets: K-EmoCon (Park et al., 2020) and ZuCo 1.0/2.0 Dataset (Hollenstein et al., 2018, 2020b). Given a succession of word-level or sentence-level EEG features and their corresponding language, Sentiment Analysis (SA) task aims to predict the sentiment label. For Relation Detection (RD), the goal is to extract semantic relations between entities in a given text. More details about the tasks, data processing, and experimental settings can be found in Appendix C.

In Table 1, we show the comparison results of the ZuCo dataset for Sentiment Analysis and Relation Detection, respectively. Our method outperforms all baselines, and the multimodal approach outperforms unimodal approaches, which further demonstrates the importance of exploring the inner

alignment between EEG and language. The results of the K-EmoCon dataset are listed in Appendix D

## 4.2 Ablation Study

To further investigate the performance of different mechanisms in the CAM, we carried out ablation experiments on the Zuco dataset, and the results are shown in Table 6 in Appendix D.2. The combination of CCA and WD performed better compared to using only one mechanism for sentiment analysis and relation detection in all model settings. We also conducted experiments on word-level, sentence-level, and concat word-level inputs, and the results are also shown in Table 6. We observe that word-level EEG features paired with their respective word generally outperform sentence-level and concat word-level in both tasks.

## 4.3 Analysis

To understand the alignment between language and EEG, we visualize the alignment weights of word-level EEG-language alignment on the ZuCo dataset. Fig. 2 and Fig. 3 show examples of negative & positive sentence word-level alignment, respectively. The sentence-level alignment visualizations are shown in Appendix D.5.

From word level alignment in Fig. 2 and 3, beta2 and gamma1 waves are most active. This is consistent with the literature, which showed that gamma waves are seen to be active in detecting emotions (Li and Lu, 2009) and beta waves have been involved in higher-order linguistic functions (e.g., discrimination of word categories). Hollenstein et al.

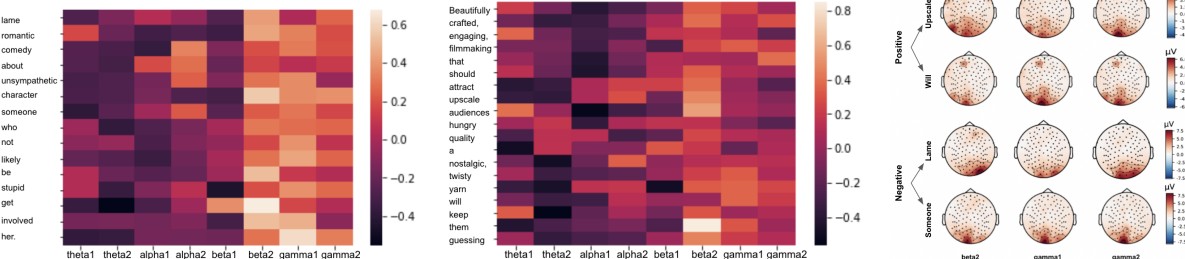

Figure 2: Negative word-level alignment. Figure 3: Positive word-level alignment. Figure 4: Brain topologies.

(2021) found that beta and theta waves were most useful in terms of model performance in sentiment analysis. In Kensinger (2009), Kensinger explained that generally, negative events are more likely to be remembered than positive events. Building off of Kensinger (2009), negative words can embed a more significant and long-lasting memory than positive words, and thus may have higher activation in the occipital and inferior parietal lobes.

We performed an analysis of which EEG feature refined the model's performance since different neurocognitive factors during language processing are associated with brain oscillations at miscellaneous frequencies. The beta and theta bands have positively contributed the most, which is due to the theta band power expected to rise with increased language processing activity and the band's relation to semantic memory retrieval (Kosch et al., 2020; Hollenstein et al., 2021). The beta's contribution can be best explained by the effect of emotional connotations of the text (Bastiaansen et al., 2005; Hollenstein et al., 2021).

In Fig. 4, we visualized the brain topologies with word-level EEG features for important and unimportant words from positive and negative sentences in the ZuCo dataset. We deemed a word important if the definition had a positive or negative connotation. 'Upscale' and 'lame' are important positive and negative words, respectively, while 'will' and 'someone' are unimportant positive and negative words, respectively. There are two areas in the brain that are heavily associated with language processing: Broca's area and Wernicke's area. Broca's area is assumed to be located in the left frontal lobe, and this region is concerned with the production of speech (Nasios et al., 2019). The left posterior superior temporal gyrus is typically assumed as Wernicke's area, and this locale is involved with the comprehension of speech (Nasios et al., 2019).

Similar to Fig. 2,3, we can observe that beta2, gamma1, and gamma2 frequency bands have the most powerful signals for all words. In Fig. 4, ac-

tivity in Wernicke's area is seen most visibly in the beta2, gamma1, and gamma2 bands for the words 'Upscale' and 'Will'. For the word 'Upscale,' we also saw activity around Broca's area for alpha1, alpha2, beta1, beta2, theta1, and theta2 bands. An interesting observation is that for the negative words, 'Lame' and 'Someone', we see very low activation in Broca's and Wernicke's areas. Instead, we see most activity in the occipital lobes and slightly over the inferior parietal lobes. The occipital lobes are noted as the visual processing area of the brain and are associated with memory formation, face recognition, distance and depth interpretation, and visuospatial perception (Rehman and Khalili, 2019). The inferior parietal lobes are generally found to be key actors in visuospatial attention and semantic memory (Numssen et al., 2021).

## 5 Conclusion

In this study, we explore the relationship between EEG and language. We propose MTAM, a Multimodal Transformer Alignment Model, to observe coordinated representations between the two modalities and employ the transformed representations for downstream applications. Our method achieved state-of-the-art performance on sentiment analysis and relation detection tasks on two public datasets, ZuCo and K-EmoCon. Furthermore, we carried out a comprehensive study to analyze the connectivity and alignment between EEG and language. We observed that the transformed features show less randomness and sparsity. The word-level language-EEG alignment clearly demonstrated the importance of the explored connectivity. We also provided brain topologies as an intuitive understanding of the corresponding activity regions in the brain, which could build the empirical neuropsychological basis for understanding the relationship between EEG and language through computational models.

## 6 Limitations

Since we proposed a new task of exploring the relationship between EEG and language, we believe there are several limitations that can be focused on in future work.

- The size of the datasets may not be large enough. Due to the difficulty and time-consumption of collecting human-related data (in addition, to privacy concerns), there are few publicly available datasets that have EEG recordings with corresponding natural language. When compared to other mature tasks, (i.e. image classification, object detection, etc), datasets that have a combination of EEG signals and different modalities are rare. In the future, we would like to collect more data on EEG signals with natural language to enhance innovation in this direction.

- The computational architecture, the MTAM model, is relatively straightforward. We agree the dual-encoder architecture is one of the standard paradigms in multimodal learning. Since our target is to explore the connectivity and relationship between EEG and language, we used a straightforward paradigm. Our model's architecture may be less complex compared to others in different tasks, such as image-text pre-training. However, we purposely avoid complicating the model's structure due to the size of the training data. We noticed when adding more layers of complexity, the model was more prone to overfitting.

- The literature lacks available published baselines. As shown in our paper, since the task is new, there are not enough published works that provide comparable baselines. We understand that the comparison is important, so we implemented several baselines by ourselves, including MLP, Bi-LSTM, Transformer, and ResNet, to provide more convincing judgment and support future work in this area.

## 7 Ethics Statement

The goal of our study is to explore the connectivity between EEG and language, which involves human subjects' data and may inflect cognition in the brain, so we would like to provide an ethics discussion.

First, all the data used in our paper are publicly available datasets: K-EmoCon and Zuco. We did not conduct any human-involved experiments by ourselves. Additionally, we do not implement any technologies on the human brain. The datasets can be found in Park et al. (2020); Hollenstein et al. (2018, 2020b)

We believe this study can empirically provide findings about the connection between natural language and the human brain. To our best knowledge, we do not foresee any harmful uses of this scientific study.

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

## A    Three paradigms of EEG and language alignment

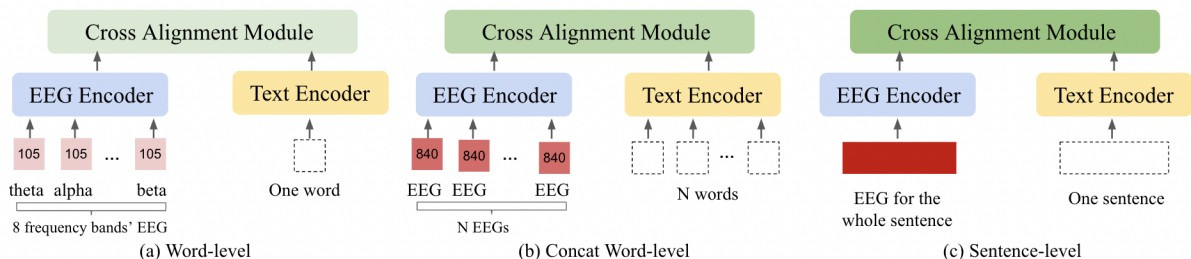

Figure 5: Three paradigms of EEG and language alignment.

## B    More Details about our Model

### B.1    Hierarchical Transformer Encoders

Let $X_e \in \mathbb{R}^{D_e}$ and $X_t \in \mathbb{R}^{D_t}$ be the two normalized input feature matrices for EEG and text, respectively, where $D_e$ and $D_t$ describes the dimensions of the feature matrices. To encode the two feature vectors, we feed them to their hierarchical transformer encoders: $V_e = E_e(X_e; W_e); V_t = E_t(X_t; W_t)$, where $E_e$ and $E_t$ denotes the separate encoders, $V_e$ and $V_t$ symbolizes the outputs for the transformed low-level features and $W_e$ and $W_t$ denotes the trainable weights for EEG and text respectively. The outputs of these two encoders can be further expanded by stating $V_e = [v_e^1, v_e^2, v_e^3, ..., v_e^n] \in \mathbb{R}^n$ and $V_t = [v_t^1, v_t^2, v_t^3, ..., v_t^k] \in \mathbb{R}^k$, where $n$ and $k$ denotes the number of instances in a given output vector and $v_e^n$ and $v_t^k$ denotes the instance itself. The details about Transformer encoders are introduced in the section below.

### B.2    Transformer Encoders

The transformer is based on the attention mechanism and outperforms previous models in accuracy and performance. The original transformer model is composed of an encoder and a decoder. The encoder maps an input sequence into a latent representation, and the decoder uses the representation along with other inputs to generate a target sequence. Our model only adopts the encoder, since we aim at learning the representations of features.

First, we feed out the input into an embedding layer, which is a learned vector representation. Then we inject positional information into the embeddings by:

$$PE_{(pos,2i)} = \sin\left(pos/10000^{2i/d_{\text{model}}}\right), \quad PE_{(pos,2i+1)} = \cos\left(pos/10000^{2i/d_{\text{model}}}\right) \qquad (1)$$

The attention model contains two sub-modules, a multi-headed attention model and a fully connected network. The multi-headed attention computes the attention weights for the input and produces an output vector with encoded information on how each feature should attend to all other features in the sequence. There are residual connections around each of the two sub-layers followed by a layer normalization, where the residual connection means adding the multi-headed attention output vector to the original positional input embedding, which helps the network train by allowing gradients to flow through the networks directly. Multi-headed attention applies a self-attention mechanism, where the input goes into three distinct fully connected layers to create the query, key, and value vectors. The output of the residual connection goes through layer normalization.

In our model, our attention model contains $N$ same layers, and each layer contains two sub-layers, which are a multi-head self-attention model and a fully connected feed-forward network. Residual connection and normalization are added in each sub-layer. So the output of the sub-layer can be expressed as: Output = LayerNorm($x$ + (SubLayer($x$))), For the Multi-head self-attention module, the attention can be expressed as: attention = Attention($Q, K, V$), where multi-head attention uses $h$ different linear transformations to project query, key, and value, which are $Q$, $K$, and $V$, respectively, and finally

concatenate different attention results:

$$\text{MultiHead(Q,K,V)} = \text{Concat}(head_1, ..., head_h)W^O \tag{2}$$

$$head_i = \text{Attention}(QW_i^Q, KW_i^K, VW_i^V) \tag{3}$$

where the projections are parameter matrices:

$$W_i^Q \in \mathbb{R}^{d_{\text{model}}\, d_k}, \quad W_i^K \in \mathbb{R}^{d_{\text{model}}\, d_k}, W_i^V \in \mathbb{R}^{d_{\text{model}}\, d_v}, \quad W_i^O \in \mathbb{R}^{hd_v \times d_{\text{model}}} \tag{4}$$

where the computation of attention adopted scaled dot-product: $\text{Attention}(Q, K, V) = \text{softmax}(\frac{QK^T}{\sqrt{d_k}})V$
For the output, we use a 1D convolutional layer and softmax layer to calculate the final output.

### B.3 Cross Alignment Module

As shown in Fig. 5, there are three paradigms of EEG and language alignment. For word level, the EEG features are divided by each word, and the objective of the alignment is to find the connectivity of different frequencies with the corresponding word. For the concat-word level, the 8 frequencies' EEG features are concatenated as a whole, and then concatenated again to match the corresponding sentence, so the alignment is to find out the relationship within the sentence. As for sentence level, the EEG features are calculated as an average over the word-level EEG features. There is no boundary for the word, so the alignment module tries to encode the embeddings as a whole, and explore the general representations. In the Cross Alignment Module (CAM), we introduced a new loss function in addition to the original cross-entropy loss. The new loss is based on Canonical Correlation Analysis (CCA) (Andrew et al., 2013) and Optimal Transport (Wasserstein Distance). As in Andrew et al. (2013), CCA aims to concurrently learn the parameters of two networks to maximize the correlation between them. Wasserstein Distance (WD), which originates from Optimal Transport (OT), has the ability to align embeddings from different domains to explore the relationship (Chen et al., 2020).

**Canonical Correlation Analysis (CCA)** is a method for exploring the relationships between two multivariate sets of variables. It learns the linear transformation of two vectors to maximize the correlation between them, which is used in many multimodal problems (Andrew et al., 2013; Qiu et al., 2018; Gong et al., 2013). In this work, we apply CCA to capture the cross-domain relationship. Let low-level transformed EEG features be $V_e$ and low-level language features be $L_t$. We assume $(V_e, V_t) \in \mathbb{R}^{n_1} \times \mathbb{R}^{n_2}$ has covariances $(\Sigma_{11}, \Sigma_{22})$ and cross-covariance $\Sigma_{12}$. CCA finds pairs of linear projections of the two views, $(w_1' V_e, w_2' V_t)$ that are maximally correlated:

$$(w_1^*, w_2^*) = \underset{w_1, w_2}{\text{argmax}}\, \text{corr}\left(w_1' V_e, w_2' V_t\right) = \underset{w_1, w_2}{\text{argmax}} \frac{w_1' \Sigma_{12} w_2}{\sqrt{w_1' \Sigma_{11} w_1 w_2' \Sigma_{22} w_2}} \tag{5}$$

In our study, we modified the structure of Andrew et al. (2013) while honoring its duty by replacing the neural networks with Transformer encoders. $w_1^*$ and $w_2^*$ denote the high-level, transformed weights from the low-level text and EEG features, respectively.

**Wasserstein Distance (WD)** is introduced in Optimal Transport (OT), which is a natural type of divergence for registration problems as it accounts for the underlying geometry of the space, and has been used for multimodal data matching and alignment tasks (Chen et al., 2020; Yuan et al., 2020; Lee et al., 2019; Demetci et al., 2020; Qiu et al., 2022; Zhu et al., 2022). In Euclidean settings, OT introduces WD $\mathcal{W}(\mu, \nu)$, which measures the minimum effort required to "displace" points across measures $\mu$ and $\nu$, where $\mu$ and $\nu$ are values observed in the empirical distribution. In our setting, we compute the temporal-pairwise Wasserstein Distance on EEG features and language features, which are $(\mu, \nu) = (V_e, V_t)$. For simplicity without loss of generality, assume $\mu \in P(\mathbb{X})$ and $\nu \in P(\mathbb{Y})$ denote the two discrete distributions, formulated as $\mu = \sum_{i=1}^n u_i \delta_{x_i}$ and $\nu = \sum_{j=1}^m v_j \delta_{y_j}$, with $\delta_x$ as the Dirac function centered on x. $\Pi(\mu, \nu)$ denotes all the joint distributions $\gamma(x, y)$, with marginals $\mu(x)$ and $\nu(y)$. The weight vectors

$u = \{u_i\}_{i=1}^n \in \Delta_n$ and $v = \{v_i\}_{i=1}^m \in \Delta_m$ belong to the $n-$ and $m-$dimensional simplex, respectively. The WD between the two discrete distributions $\mu$ and $\nu$ is defined as:

$$\mathcal{WD}(\mu, \nu) = \inf_{\gamma \in \Pi(\mu,\nu)} \mathbb{E}_{(x,y)\sim\gamma}[c(x,y)] = \min_{\mathbf{T} \in \Pi(\mathbf{u},\mathbf{v})} \sum_{i=1}^n \sum_{j=1}^m T_{ij} \cdot c(x_i, y_j) \qquad (6)$$

where $\Pi(u,v) = \{T \in \mathbb{R}_+^{n \times m} | T\mathbf{1}_m = u, T^\top \mathbf{1}_n = v\}$, $\mathbf{1}_n$ denotes an $n-$dimensional all-one vector, and $c(x_i, y_j)$ is the cost function evaluating the distance between $x_i$ and $y_j$.

**Loss Objective**    The loss objective for the CAM module can be formalized as: $Loss = l_{CE} + \alpha_1 l_{CCA} + \alpha_2 l_{WD}$, where $\alpha_i \in \{0, 1\}, i \in (1, 2)$ controls the weights of different parts of alignment-based loss objective.

## C    Experimental Settings

### C.1    Downstream Tasks

In this study, we evaluate our method on two downstream tasks: Sentiment Analysis (SA) and Relation Detection (RD) of two datasets: K-EmoCon (Park et al., 2020) and ZuCo 1.0/2.0 Dataset (Hollenstein et al., 2018, 2020b).

**Sentiment Analysis (SA)**    Given a succession of word-level or sentence-level EEG features and their corresponding language, the task is to predict the sentiment label. The ZuCo 1.0 dataset consists of sentences from the Stanford Sentiment Treebank, which contains movie reviews and their corresponding sentiment label (i.e., positive, neutral, negative) (Socher et al., 2013). The K-EmoCon dataset categorizes emotion annotations as valence, arousal, happy, sad, nervous, and angry. For each emotion, the participant labeled the extent of the given emotion felt by following a Likert-scale paradigm. Arousal and valence are rated 1 to 5 (1: very low; 5: very high). Happy, sad, nervous, and angry emotions are rated 1 to 4, where 1 means very low and 4 means very high. The ratings are dominantly labeled as very low and neutral. Therefore to combat class imbalance, we collapse the labels to binary and ternary settings.

**Relation Detection (RD)**    The goal of relation detection (also known as relation extraction or entity association) is to extract semantic relations between entities in a given text. For example, in the sentence, "June Huh won the 2022 Fields Medal.", the relation *AWARD* connects the two entities "June Huh" and "Fields Medal" together. The ZuCo 1.0/2.0 datasets provide the ground truth labels and texts for this task. We use texts from the Wikipedia relation extraction dataset (Culotta et al., 2006) that has 10 relation categories: award, control, education, employer, founder, job title, nationality, political affiliation, visited, and wife (Hollenstein et al., 2018, 2020b).

### C.2    Datasets and Data Processing

**K-EmoCon Dataset**    K-EmoCon (Park et al., 2020) is a multimodal dataset including videos, speech audio, accelerometer, and physiological signals during a naturalistic conversation. After the conversation, each participant watched a recording of themselves and annotated their own and partner's emotions. Five external annotators were recruited to annotate both parties' emotions, six emotions in total (Arousal, Valence, Happy, Sad, Angry, Nervous). The NeuroSky MindWave headset captured EEG signals from the left prefrontal lob (FP1) at a sampling rate of 125 Hz in 8 frequency bands: delta (0.5–2.75Hz), theta (3.5–6.75Hz), low-alpha (7.5–9.25Hz), high-alpha (10–11.75Hz), low-beta (13–16.75Hz), high-beta (18–29.75Hz), low-gamma (31–39.75Hz), and middle-gamma (41–49.75Hz). We used Google Cloud's Speech-to-Text API to transcribe the audio data into text.

**ZuCo Dataset**    The ZuCo Dataset (Hollenstein et al., 2018, 2020b) is a corpus of EEG signals and eye-tracking data during natural reading. The tasks during natural reading can be separated into three categories: sentiment analysis, natural reading, and task-specified reading. During sentiment analysis, the participant was presented with 400 positive, neutral, and negative labeled sentences from the Stanford Sentiment Treebank (Socher et al., 2013). The EEG data used in this study can be categorized into

sentence-level and word-level features. The sentence-level features are the averaged word-level EEG features for the entire sentence duration. The word-level EEG features are for the first fixation duration (FFD) of a specific word, meaning when the participant's eye met the word, the EEG signals were recorded. For both word and sentence-level features, 8 frequency bands were recorded at a sampling frequency of 500 Hz and denoted as the following: theta1 (4-6Hz), theta2 (6.5–8Hz), alpha1 (8.5–10Hz), alpha2 (10.5–13Hz), beta1 (13.5–18Hz), beta2 (18.5–30Hz), gamma1 (30.5–40Hz), and gamma2 (40–49.5Hz).

## C.3  Experimental Setup

The hierarchical transformer encoders follow the standard skeleton from Vaswani et al. (2017), excluding its complexity. To avoid overfitting, we adopt the oversampling strategy for data augmentation (Hübschle-Schneider and Sanders, 2019), which ensures a balanced distribution of classes included in each batch. The train/test/validation splitting is $(80\%, 10\%, 10\%)$ as in Hollenstein et al. (2021). The EEG features are extracted from the datasets in 8 frequency bands and normalized with Z-score according to previous work (S. Yousif et al., 2020; Fdez et al., 2021; Du et al., 2022) over each frequency band. To preserve relatability, the word and sentence embeddings are also normalized with Z-scores. We use pre-trained language models to generate text features (Devlin et al., 2019), where all texts are tokenized and embedded using the BERT-uncased-base model. Each sentence has an average length of 20 tokens, so we instantiate a max length of 32 with padding. In the case of word-level, we use an average length of 4 tokens for each word and establish a max length of 10 with padding. The token vectors' from the four last hidden layers of the pre-trained model are withdrawn and averaged to get a final sentence or word embedding. These embeddings are used during the sentence-level and word-level settings. For concat word-level, we simply concatenate the word embeddings for their respective sentence. All the experimental parameters are listed in Appendix C.4.

## C.4  Experiment Parameters, Code and Dataset

Our model's parameters used in the experiments are listed in Table 2. Parameters with the best performance are marked in bold. Our anonymous code is available at `https://anonymous.4open.science/r/EMNLP_2023-B08D/`.

Table 2: Experiment parameters used in the paper, where the best ones are marked in bold

| Task | Batch Size | Encoder Layers | Att. Heads | In Channel Size | Out Channel Size |
|---|---|---|---|---|---|
| Sentiment Analysis | [8, 16, 32, **64**, 128] | [**1**, 2, 4, 6, 12] | [1, 2, **3**, 4, 5, 6, 7, 8, 12] | [8, **16**, 32, 64, 128, 256] | [8, 16, **32**, 64, 128, 256] |
| Relation Detection | [8, 16, 32, **64**, 128] | [**1**, 2, 4, 6, 12] | [1, 2, **3**, 4, 5, 6, 7, 8, 12] | [8, **16**, 32, 64, 128, 256] | [8, 16, **32**, 64, 128, 256] |

| Task | Kernel Sizes | Dropout | Epochs | Warmup Steps |
|---|---|---|---|---|
| Sentiment Analysis | [1, **3**, 5] | [0.1, 0.2, **0.3**, 0.4, 0.5, 0.6, 0.7] | [10, 20, 50, 100, **200**] | [1000, **2000**, 4000] |
| Relation Detection | [1, **3**, 5] | [0.1, 0.2, **0.3**, 0.4, 0.5, 0.6, 0.7] | [10, 20, 50, 100, **200**] | [1000, **2000**, 4000] |

## C.5  Baselines

The area of multimodal learning of EEG and language is not well explored, and to the best of our knowledge, only Hollenstein et al. (2021)'s approach was directly comparable to our study. However, to make a fair evaluation, we implemented the following state-of-the-art representative approaches as baselines for verification: MLP (Ruppert, 2004), Bi-LSTM (Graves and Schmidhuber, 2005; Zhou et al., 2016), Transformer (Vaswani et al., 2017), and ResNet (He et al., 2016).

In this section, we present implementation details for our multilayer perceptron (MLP), ResNet, and BiLSTM models during baseline retrieval. Throughout all baseline results, we used a pre-trained BERT-uncased-base model to extract useful features for text. In the case of EEG features, we used the signals as is. Both text and EEG features were normalized with a Z-score before inputting them into the models. We also used the cross-entropy loss function for all baseline results. We configure the MLP with 6 hidden layers. At every step before the last output layer, we established a rectified linear unit activation function and a dropout rate of 0.3. Starting from the input layer, we use a hidden layer sizes of 256, 128, and 64 for our baseline results. Our 1D ResNet architecture has 34 layers (Hong et al., 2020). The BiLSTM

model has 4 layers with a size of 128 and 64, respectively. Once the initial embedding is fed into the BiLSTM model, we use a pack padded sequence function to ignore the padded elements. The comparison of implementation details of baseline methods and our methods is shown in Table 3

Table 3: Implementation details for the baselines models and the comparison with our methods.

| Model | Architecture | | Parameters | |
|---|---|---|---|---|
| | Num. of Layers | Size of Layers | Dropout | Batch Size |
| MLP-EEG | [2, 4, **6**] | [64, 128, 256] | [0.1, **0.3**, 0.5] | [8, 16, **32**] |
| MLP-Text | [2, **4**, 6] | [64, 128, 256] | [0.1, **0.3**, 0.5] | [8, 16, **32**] |
| BiLSTM-EEG | [2, 3, **4**] | [32, 64, 128, 256] | [0.1, **0.3**, 0.5] | [8, 16, **32**] |
| BiLSTM-Text | [2, 3, **4**] | [32, 64, 128, 256] | [0.1, **0.3**, 0.5] | [8, 16, **32**] |
| Transformer-EEG | [1, **2**, 4] | [8, 16, 32, 64] | [0.1, **0.3**, 0.5] | [8, 16, 32, **64**] |
| Transformer-Text | [**1**, 2, 4] | [8, 16, 32, 64] | [0.1, **0.3**, 0.5] | [8, 16, 32, **64**] |
| ResNet-EEG | [**34**, 50, 96] | [32, 64, 128, 256] | [0.1, **0.3**, 0.5] | [8, 16, **32**] |
| ResNet-Text | [**34**, 50, 96] | [32, 64, 128, 256] | [0.1, **0.3**, 0.5] | [8, 16, **32**] |
| Ours-EEG | [1, **2**, 4] | [8, 16, 32, 64] | [0.1, **0.3**, 0.5] | [8, 16, **32**] |
| Ours-Text | [**1**, 2, 4] | [8, 16, 32, 64] | [0.1, **0.3**, 0.5] | [8, 16, **32**] |
| Ours-Multimodal | [1, **2**, 4] | [8, 16, 32, 64] | [0.1, **0.3**, 0.5] | [8, 16, 32, **64**] |

## D  Additional Experimental Results

### D.1  Experimental Results on K-EmoCon dataset

To the best of our knowledge, there is no existing work where EEG or text is used for the K-EmoCon dataset. However, other modalities such as audio, video, blood volume pulse (BVP), electrodermal activity (EDA), body temperature (TEMP), skin temperature (SKT), accelerometer (ACC) and heart rate (HR) have been used to perform sentiment analysis. As shown in Table 4, our model outperforms previous method, with even less domains' data, showing the connectivity between EEG and language and also the advantages of exploring them for downstream applications.

Table 4: Comparison of performance on K-EmoCon dataset with different physiological signals as inputs on the Sentiment Analysis task.

| Model | Modalities | Rec | F1 | Acc |
|---|---|---|---|---|
| CNN + Transformer (Quan et al., 2021) | Video and Audio | 0.693 | 0.712 | 0.725 |
| CNN Fusion (Dissanayake et al., 2022) | ACC, BVP, EDA, TEMP | NA | 0.562 | 0.591 |
| Convolution-augmented Transformer (Yang et al., 2022) | BCP, EDA, HR, SKT | 0.655 | 0.564 | NA |
| Transformer (Yang et al., 2022) | BCP, EDA, HR, SKT | 0.628 | 0.518 | NA |
| BiLSTM (Yang et al., 2022) | BCP, EDA, HR, SKT | 0.563 | 0.473 | NA |
| Ours | Text, EEG | **0.720** | **0.729** | **0.733** |

Table 5: Comparison with baselines on K-EmoCon dataset for Sentiment Analysis.

| Model | Prec | Rec | F1 | Acc |
|---|---|---|---|---|
| MLP-EEG | 0.295 | 0.317 | 0.222 | 0.231 |
| MLP-Text | 0.263 | 0.272 | 0.182 | 0.180 |
| Bi-LSTM-EEG | 0.340 | 0.354 | 0.226 | 0.220 |
| Bi-LSTM-Text | 0.241 | 0.329 | 0.125 | 0.224 |
| Transformer-EEG | 0.399 | 0.411 | 0.405 | 0.484 |
| Transformer-Text | 0.454 | 0.492 | 0.472 | 0.443 |
| ResNet-EEG | 0.456 | 0.389 | 0.202 | 0.229 |
| ResNet-Text | 0.133 | 0.348 | 0.169 | 0.224 |
| Ours-EEG | **0.591** | **0.516** | **0.551** | **0.591** |
| Ours-Text | **0.524** | **0.561** | **0.509** | **0.542** |
| Ours-Multimodal | **0.739** | **0.720** | **0.729** | **0.733** |

In Table 5, we show the comparison results of different methods on the K-EmoCon dataset. From Table 5, we can see that our method outperforms the other baselines, and the multimodal approach outperforms the unimodal approach, which also demonstrates the effectiveness of our method.

## D.2 Ablation results on the components in the CAM module

To further investigate the performance of different mechanisms in the CAM, we carried out ablation experiments on the Zuco dataset, and the results are shown in Table 6 in Appendix D.2. The combination of CCA and WD performed better compared to using only one mechanism for sentiment analysis and relation detection in all model settings. We also conducted experiments on word-level, sentence-level, and concat word-level inputs, and the results are also shown in Table 6. We observe that word-level EEG features paired with their respective word generally outperform sentence-level and concat word-level in both tasks.

Table 6: Ablation results on the components in the CAM module (best results in bold).

| Dataset | Model | Sentence Level | | | | Word Level | | | | Concat Word Level | | | |
|---|---|---|---|---|---|---|---|---|---|---|---|---|---|
| | | Prec | Rec | F1 | Acc | Prec | Rec | F1 | Acc | Prec | Rec | F1 | Acc |
| ZuCo (SA) | Ours-CCA-Text | 0.748 | 0.746 | 0.747 | 0.707 | 0.701 | 0.733 | 0.717 | 0.769 | 0.752 | 0.787 | 0.769 | 0.744 |
| | Ours-CCA-EEG | 0.758 | 0.763 | 0.761 | 0.758 | 0.761 | 0.763 | 0.761 | 0.758 | 0.748 | 0.751 | 0.749 | 0.748 |
| | Ours-CCA-All | 0.790 | 0.765 | 0.777 | 0.793 | 0.791 | 0.783 | 0.787 | 0.793 | 0.767 | 0.778 | 0.773 | 0.778 |
| | Ours-WD-Text | 0.718 | 0.704 | 0.711 | 0.724 | 0.753 | 0.747 | 0.750 | 0.770 | 0.740 | 0.731 | 0.735 | 0.733 |
| | Ours-WD-EEG | 0.772 | 0.744 | 0.754 | 0.758 | 0.786 | 0.799 | 0.792 | 0.793 | 0.710 | 0.706 | 0.708 | 0.694 |
| | Ours-WD-All | 0.774 | 0.733 | 0.752 | 0.804 | 0.817 | 0.809 | 0.803 | 0.805 | 0.781 | 0.784 | **0.837** | 0.780 |
| | Ours-CCA+WD-Text | 0.763 | 0.762 | 0.762 | 0.735 | 0.747 | 0.749 | 0.748 | 0.753 | 0.720 | 0.744 | 0.723 | 0.742 |
| | Ours-CCA+WD-EEG | 0.779 | 0.783 | 0.781 | 0.779 | 0.783 | 0.786 | 0.784 | 0.781 | 0.773 | 0.770 | 0.771 | 0.772 |
| | Ours-CCA+WD-All | **0.822** | **0.829** | **0.826** | **0.826** | **0.821** | **0.812** | **0.816** | **0.827** | **0.802** | **0.809** | 0.806 | **0.813** |
| ZuCo (RD) | Ours-CCA-Text | 0.525 | 0.524 | 0.524 | 0.502 | 0.489 | 0.463 | 0.475 | 0.455 | 0.386 | 0.459 | 0.419 | 0.421 |
| | Ours-CCA-EEG | 0.596 | 0.648 | 0.612 | 0.604 | 0.546 | 0.547 | 0.546 | 0.547 | 0.520 | 0.561 | 0.540 | 0.525 |
| | Ours-CCA-All | 0.624 | 0.651 | 0.620 | 0.617 | 0.638 | 0.646 | 0.642 | 0.633 | 0.596 | 0.606 | 0.600 | 0.599 |
| | Ours-WD-Text | 0.539 | 0.514 | 0.526 | 0.534 | 0.537 | 0.499 | 0.518 | 0.521 | 0.478 | 0.462 | 0.470 | 0.479 |
| | Ours-WD-EEG | 0.626 | 0.625 | 0.625 | 0.610 | 0.647 | 0.653 | 0.650 | 0.648 | 0.642 | 0.666 | 0.653 | 0.648 |
| | Ours-WD-All | 0.642 | 0.686 | 0.663 | 0.704 | 0.718 | **0.754** | 0.736 | 0.731 | 0.718 | 0.693 | 0.706 | 0.733 |
| | Ours-CCA+WD-Text | 0.534 | 0.538 | 0.536 | 0.526 | 0.534 | 0.565 | 0.552 | 0.549 | 0.455 | 0.494 | 0.472 | 0.473 |
| | Ours-CCA+WD-EEG | 0.676 | 0.698 | 0.687 | 0.673 | 0.669 | 0.681 | 0.678 | 0.679 | 0.658 | 0.690 | 0.669 | 0.669 |
| | Ours-CCA+WD-All | **0.742** | **0.746** | **0.742** | **0.744** | **0.741** | 0.743 | **0.742** | **0.746** | **0.737** | **0.734** | **0.736** | **0.737** |

## D.3 Full Brain Topological Maps

In the paper, we only showed the brain topological maps for three frequency bands due to page limit, here we provide full brain topological maps for all the eight frequency bands, and the results are shown in Figure 6. We can observe the beta2, gamma1, and gamma2 frequency bands having the most powerful signals for all words.

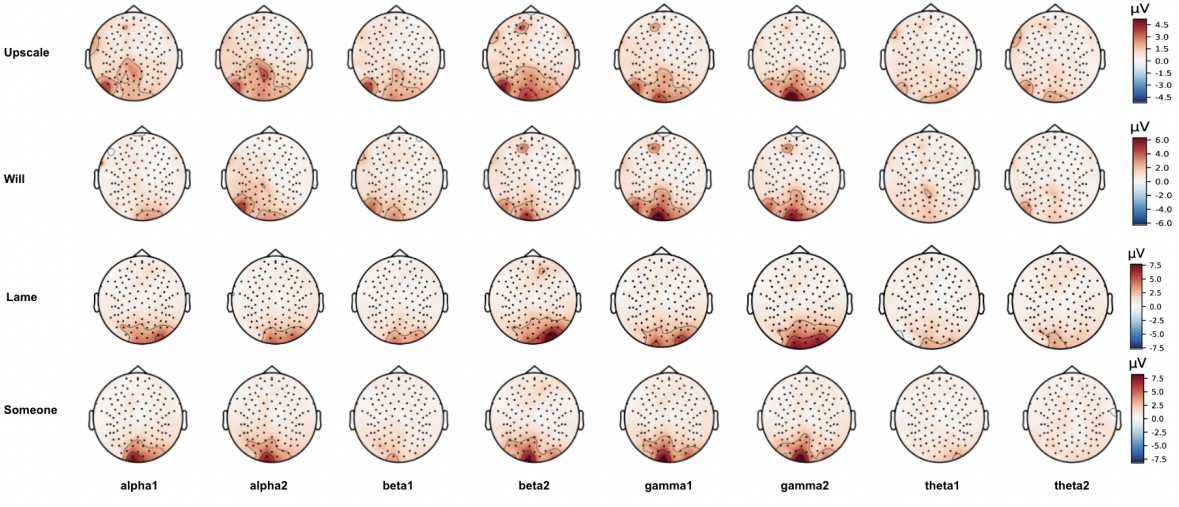

Figure 6: Positive and Negative word brain topologies (Sentiment Analysis)

### D.4 t-SNE Feature Projections

In order to interpret the performance improvement, we visualized the original feature distribution and the transformed feature distribution. As shown in Fig. 7, the transformed feature distribution makes better clusters than the original one. The features learned by CAM can be more easily separable, showing the effectiveness of discovering and encoding the relationship between EEG and language. Figures 8,9,10 show more t-SNE projection results of the K-EmoCon dataset on Sentiment Analysis task.

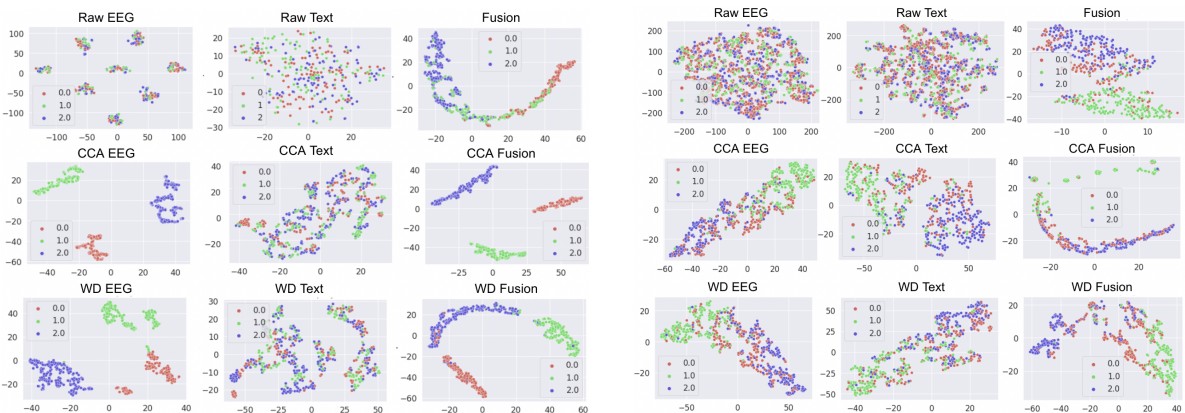

Figure 7: TSNE projection comparison of untransformed & transformed features of ZuCo dataset, where different colors represent different classes.

Figure 8: Transformed feature projections of K-EmoCon dataset on Sentiment Analysis, where different colors represent different classes.

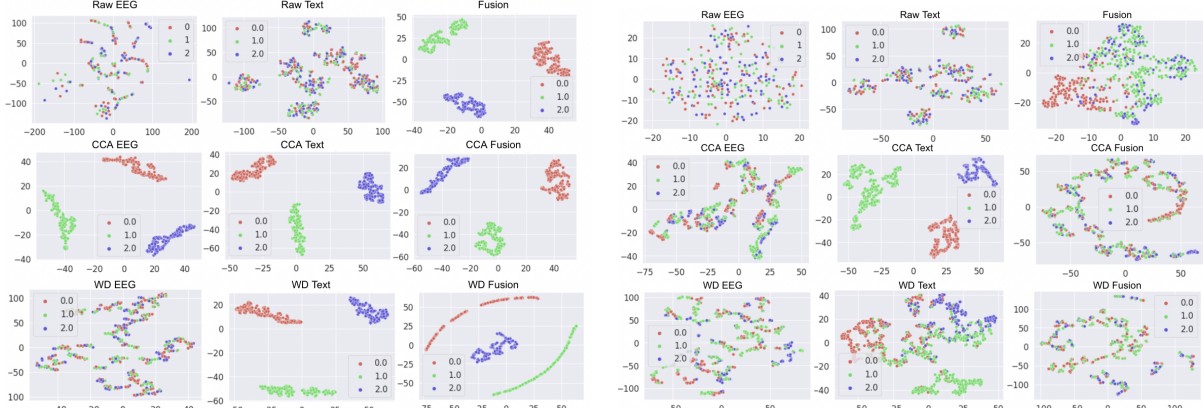

Figure 9: Transformed feature projections of ZuCo dataset on Sentiment Analysis, word-level, where different colors represent different classes.

Figure 10: Transformed feature projections of ZuCo dataset on Sentiment Analysis, concat word-level, where different colors represent different classes.

### D.5 Sentence-level Alignment

Figure 11 shows the negative and positive sentence-level alignment weights of ZuCo dataset. In Figure 11, we can find that alpha1, beta1,and gamma1 frequency bands show larger different response between negative and positive sentences.

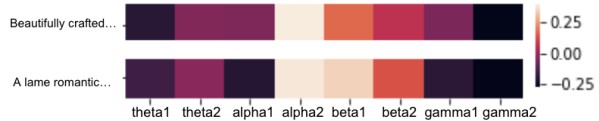

Figure 11: Negative and Positive sentence-level alignment of ZuCo dataset.

### D.6 Baseline Results

In this section, we provided baseline results that directly used either EEG, language, or fusion as input for the downstream applications. The results are shown in Table 7 and Table 8.

Table 7: Baseline results

| Dataset | Model | Task | Sentence Level | | | | Word Level | | | | Concat Word Level | | | |
|---|---|---|---|---|---|---|---|---|---|---|---|---|---|---|
| | | | Prec | Rec | F1 | Acc | Prec | Rec | F1 | Acc | Prec | Rec | F1 | Acc |
| ZuCo | Transformer-Text | Sentiment Analysis, Ternary | 0.548 | 0.546 | 0.547 | 0.507 | 0.527 | 0.533 | 0.530 | 0.582 | 0.558 | 0.547 | 0.552 | 0.550 |
| | Transformer-EEG | Sentiment Analysis, Ternary | 0.665 | 0.659 | 0.662 | 0.662 | 0.624 | 0.630 | 0.627 | 0.664 | 0.624 | 0.630 | 0.627 | 0.614 |
| | Transformer-Text | Relation Detection | 0.428 | 0.487 | 0.444 | 0.420 | 0.488 | 0.491 | 0.489 | 0.487 | 0.301 | 0.299 | 0.300 | 0.300 |
| | Transformer-EEG | Relation Detection | 0.502 | 0.440 | 0.468 | 0.479 | 0.339 | 0.341 | 0.340 | 0.358 | 0.310 | 0.316 | 0.313 | 0.315 |
| K-EmoCon | Transformer-Text | Sentiment Analysis, Ternary | 0.454 | 0.492 | 0.472 | 0.443 | - | - | - | - | - | - | - | - |
| | Transformer-EEG | Sentient Analysis, Ternary | 0.399 | 0.411 | 0.405 | 0.484 | - | - | - | - | - | - | - | - |
| | Transformer-Text | Sentiment Analysis, Binary | 0.597 | 0.557 | 0.508 | 0.657 | - | - | - | - | - | - | - | - |
| | Transformer-EEG | Sentient Analysis, Binary | 0.814 | 0.877 | 0.808 | 0.890 | - | - | - | - | - | - | - | - |

Table 8: Baseline fusion results.

| Dataset | Model | Task | Sentence Level | | | | Word Level | | | | Concat Word Level | | | |
|---|---|---|---|---|---|---|---|---|---|---|---|---|---|---|
| | | | Prec | Rec | F1 | Acc | Prec | Rec | F1 | Acc | Prec | Rec | F1 | Acc |
| ZuCo | Fusion | Sentiment Analysis, Ternary | 0.674 | 0.686 | 0.680 | 0.677 | 0.706 | 0.707 | 0.707 | 0.706 | 0.654 | 0.671 | 0.662 | 0.674 |
| | Fusion | Relation Detection | 0.552 | 0.531 | 0.541 | 0.530 | 0.491 | 0.487 | 0.489 | 0.488 | 0.398 | 0.402 | 0.400 | 0.400 |
| K-EmoCon | Fusion | Sentiment Analysis, Ternary | 0.512 | 0.520 | 0.516 | 0.515 | - | - | - | - | - | - | - | - |
| | Fusion | Sentiment Analysis, Binary | 0.982 | 0.933 | 0.957 | 0.918 | - | - | - | - | - | - | - | - |

# E More Related Work

**Multimodal Learning of EEG and Other Domains** EEG signal is a popular choice as a modality in multimodal learning. Ben Said et al. (2017) used EMG signals jointly with EEG in a bi-autoencoder architecture and increased accuracies for sentiment analysis. Bashar (2018) integrated ECG and EEG signals in a human identification task, where fused classifiers produced the highest score. Liu et al. (2019a, 2022); Bao et al. (2019) extracted correlated features between EEG and eye movement data for emotion classification, showing transformed features are more homogeneous and discriminative. Guo et al. (2019) collected eye images, eye movement data, and EEG signals, and encoded the three modalities with a bimodal deep autoencoder. Ortega and Faisal (2021) fed fNIRS and EEG to decode bimanual grip force and resulted in increased performance, compared to single modality models. There are also efforts to find correlations between EEG and visual stimulus frequencies (Saeidi et al., 2021). A common theme occurring among these works showed EEG paired with other domains could boost performance.

**Multimodal Learning of Language and Other Brain Signals** Recently, language and cognitive data were also used together in multimodal settings to complete desirable tasks (Wang and Ji, 2021; Hollenstein et al., 2019, 2021, 2020a). Wehbe et al. (2014) used a recurrent neural network to perform word alignment between MEG activity and the generated word embeddings. Toneva and Wehbe (2019) utilized word-level MEG and fMRI recordings to compare word embeddings from large language models. Schwartz et al. (2019) used MEG and fMRI data to fine-tune a BERT language model (Devlin et al., 2019) and found that the relationships between these two modalities were generalized across participants. Huang et al. (2020) leveraged CT images and text from electronic health records to classify pulmonary embolism cases and observed that the multimodal model with late fusion achieved the best performance. Murphy et al. (2022) found semantic categories between MEG and language. However, the relationship between language and EEG has not been explored before.

**Multimodal Learning of EEG and Language** Hale et al. (2019) related EEG signals to the states of a neural phrase structure parser and showed that through EEG signals, models were correlating syntactic properties to a specific genre of text. Foster et al. (2021) applied EEG signals to predict specific values of each dimension in a word vector through regression models. Wang and Ji (2021) used word-level EEG features to decode corresponding text tokens through an open vocabulary, sequence-to-sequence framework. Hollenstein et al. (2021) focused on a multimodal approach by utilizing a combination of EEG, eye-tracking, and text data to improve NLP tasks. They used a variation of LSTM and CNN to decode the EEG features, but did not explore the relationship between EEG and language. Their proposed multimodal framework follows the bi-encoder approach (Choi et al., 2021) where the two modalities are encoded separately (Hollenstein et al., 2021).