# OpenReview forum: "Can Brain Signals Reveal Inner Alignment with Human Languages?"
_EMNLP/2023/Conference — EMNLP 2023 Findings_

### Official Review · Reviewer_HtV4 · 2023-07-31

**Typos Grammar Style And Presentation Improvements:** See Reasons To Reject.
**Soundness:** 3

**Excitement:**

3: Ambivalent: It has merits (e.g., it reports state-of-the-art results, the idea is nice), but there are key weaknesses (e.g., it describes incremental work), and it can significantly benefit from another round of revision. However, I won't object to accepting it if my co-reviewers champion it.

**Paper Topic And Main Contributions:**

This paper explores the relationship and dependency between EEG and languages. Author suggests a Multimodal Transformer Alignment Model (MTAM) which basically consists of dual-encoder with cross-alignment between EEG and languages. Also, canonical correlation analysis (CCA) and Wasserstein distance (WD) are considered as alignment mechanism where the combined one works the best. Two different downstream tasks are covered to reveal the novelty of work. Also, the authors share the publicly available software.

**Questions For The Authors:**

Here, I copy and past the questions covered in Reasons To Reject to make authors to readily state the answers.

Q1. Language is discrete data and EEG is continuous data. How did you explain this difference to align between them?

Q2. EEG is highly random in nature and very subjective between people which leads to hard to extract the consistent features. How did you deal with this problem?

Q3. Why the authors consider CCA and WD? How about KL divergence or Euclidean distance?

Q4. How did you combine CCA and WD? This explanation should be included in main paper, not Appendix.

**Reasons To Accept:**

- Initial work to explore the fundamental relationship and connectivity between EEG and language through computational multimodal methods.

- Suggest the simple dual-encoder model (MTAM) which reveals great performances in two different tasks.

**Reasons To Reject:**

- This paper should explain two high-level questions before suggesting author's approach (at least literature works should be covered).
(1) Language is discrete data and EEG is continuous data. How did you explain this difference to align between them?
(2) EEG is highly random in nature and very subjective between people which leads to hard to extract the consistent features. How did you deal with this problem?

- This is NLP conference which means many people are not familiar with the concept of physiological signals. The authors need to kindly explain the concepts in the work.
(1) Explain a difference between EEG and MEG.
(2) Explain more details about the used datasets.

- The above two reasons suggest the author to prepare a long paper to cover all the details. For example,
(1) Why the authors consider CCA and WD? How about KL divergence or Euclidean distance?
(2) How did you combine CCA and WD? This explanation should be included in main paper, not Appendix.

- This paper needs to be edited/reviewed again by authors before submission since there are many mistakes. Few of them are shared here.
(1) Please explain what MEG means (line 105).
(2) were generalized -> which were generalized (line 112).
(3) several -> two (line165)
(4) Appendix ?? -> Appendix D.2 (line 197).
(5) Correct quotation marks (line 249-251).

**Reproducibility:**

4: Could mostly reproduce the results, but there may be some variation because of sample variance or minor variations in their interpretation of the protocol or method.

**Reviewer Confidence:**

4: Quite sure. I tried to check the important points carefully. It's unlikely, though conceivable, that I missed something that should affect my ratings.

---

> ### Author Rebuttal · Authors · 2023-08-28
>
> Thank you for your valuable feedback! We appreciate it for recognizing our work “Initial work to explore the fundamental relationship and connectivity between EEG and language through computational multimodal methods”. Please find our detailed response to your concerns outlined below.
>
> 1. ***Language is discrete data and EEG is continuous data. How did you explain this difference to align between them?***
>
> - Thank you for the question.  We incorporated this component in both Appendix Section A and Section B.3, as a result of page limit. Illustrated in Figure 5 within the Appendix are three EEG and language alignment paradigms: word level, concatenated word level, and sentence-level.
> - For the word level approach, the EEG features are segregated on a per-word basis, aiming to establish connections between different frequencies and their corresponding words. On the other hand, in the concat-word level approach, the EEG features from the 8 frequencies are combined into a unified entity, then concatenated again to correspond with the sentence. This alignment strategy seeks to uncover relationships within the sentence structure. For the sentence level perspective, the EEG features are computed as an average across the word-level EEG features. In this context, the alignment module operates without word boundaries, endeavoring to encode the embeddings as a cohesive entity, thereby delving into overarching representations.
>
> 2. ***EEG is highly random in nature and very subjective between people which leads to hard to extract the consistent features. How did you deal with this problem?***
>
>
> - Regarding the K-Emocon dataset, it's worth noting that the authors of the original work did not disclose the precise preprocessing techniques applied to the EEG waves due to privacy concerns extending beyond the scope of shared information [1]. However, we are aware that the EEG signals were captured using the Neurosky device. This device inherently conducts internal preprocessing on raw EEG signals to derive attention and meditation levels, along with brainwave band information. Furthermore, the authors incorporated a Fast Fourier Transform analysis to the EEG signals. In our approach, we followed a similar path. We began by normalizing the amplitudes of the EEG signals using the Z-score technique for each frequency. This normalization process was adopted from previous research [2, 3, 4].
> - In the case of the ZuCo Dataset, the original authors performed a comprehensive preprocessing of the EEG signals utilizing various tools and techniques. Notably, they employed Automagic [5] for initial preprocessing steps. To ensure data quality, they detected and removed problematic electrodes that exhibited noise, flatline, or low-frequency signals. The clean_rawdata function from the EEGLab library was used for this purpose. Further processing included high-pass filtering the EEG signals at 0.5 Hz and applying a notch filter in the range of 49-51 Hz to eliminate power line interference. Eye-related artifacts were addressed by regression of the EOG channels. The authors also integrated the MARA tool to automate the rejection of artifacts, and any electrodes presenting issues were interpolated. A critical alignment step was carried out by synchronizing the EEG data with eye-tracking information using the "EYE EEG extension," a process crucial for precise temporal alignment. For a more in-depth analysis, the EEG signals underwent frequency band filtering, followed by a Hilbert transform to derive power dynamics. A notable analysis point was the comparison of power between frontal left and right electrodes. Stringent criteria were established to identify and exclude artifacts exceeding ±90μV. Following these steps, akin to the K-EmoCon dataset and prior research [2, 3, 4], we proceeded to normalize the preprocessed EEG signals using the Z-score method.
>
> > [1] Park, Cheul Young, et al. “K-EmoCon, a Multimodal Sensor Dataset for Continuous Emotion Recognition in Naturalistic Conversations.” Scientific Data, vol. 7, no. 1, 8 Sept. 2020, https://doi.org/10.1038/s41597-020-00630-y.
> > [2] Du, Yang, et al. “EEG Temporal–Spatial Transformer for Person Identification.” Scientific Reports, vol. 12, no. 1, 23 Aug. 2022, https://doi.org/10.1038/s41598-022-18502-3.
> > [3] Fdez, Javier, et al. “Cross-Subject EEG-Based Emotion Recognition through Neural Networks with Stratified Normalization.” Frontiers in Neuroscience, vol. 15, 3 Feb. 2021, https://doi.org/10.3389/fnins.2021.626277. Accessed 27 Aug. 2021.
> > [4] S. Yousif, Enas, et al. “Electroencephalogram Signals Classification Based on Feature Normalization.” IOP Conference Series: Materials Science and Engineering, vol. 928, no. 3, 1 Nov. 2020, p. 032028, https://doi.org/10.1088/1757-899x/928/3/032028.
> > [5] Automagic [https://github.com/methlabUZH/automagic](https://github.com/methlabUZH/automagic)
>
> 3.  ***Why the authors consider CCA and WD? How about KL divergence or Euclidean distance?***
>
> - Thank you for the question.
> - WD can be conceptualized as a metric representing the minimal "cost" required to transform one distribution of points into another. Conversely, CCA identifies a collection of canonical variates – which are orthogonal linear combinations of variables within each set – that optimally elucidate variability within and between sets. In broad terms, WD primarily emphasizes disparities in distributions, whereas CCA centers on canonical variables, thus exhibiting synergistic benefits when employed in conjunction.
> - We experimented with the Euclidean distance; however, the outcomes were unsatisfactory. Consequently, we chose not to include these results in the paper.
> - When juxtaposed with WD and CCA, KL divergence might introduce added complexity. Given that we've already incorporated WD and CCA as two complementary loss objectives within our existing framework, we decided against introducing a third loss. However, we acknowledge the potential of KL divergence and plan to explore its applicability in our future research.
>
> 4. ***How did you combine CCA and WD? This explanation should be included in main paper, not Appendix.***
>
> Thank you for the question. We detailed the entire loss objective, which outlines the integration of CCA and WD, in Appendix pages L750-L752, constrained by page limitations. We will add it to the main paper in our revision. Thank you very much for the suggestion.
>
>
> 5. ***More introduction of physiological signals***
>
> Thank you for the suggestion! In our revision, we will expand on the introduction of the concept of physiological signals, encompassing distinctions between EEG and MEG, as well as providing additional details about the datasets employed.
>
> 6. ***Typos in writing***
>
> Thank you so much! We will correct them in our revision!
>
> 7. ***Reproducibility***
>
> We would respectfully inquire whether the reviewer identifies any aspects they believe are insufficient for reproducing our work. The code and data necessary for replication can be accessed through an anonymous link: [https://anonymous.4open.science/r/EMNLP_2023-B08D/](https://anonymous.4open.science/r/EMNLP_2023-B08D/) (mentioned in paper lines L033-L034). This link encompasses the complete setup, including environment configuration, raw and processed data, along with scripts for processing, training, testing, and visualization. We are confident that these resources facilitate result reproduction. If there are any additional materials you require, please do not hesitate to inform us.

---

### Official Review · Reviewer_VHS7 · 2023-08-05

**Soundness:** 3

**Excitement:**

3: Ambivalent: It has merits (e.g., it reports state-of-the-art results, the idea is nice), but there are key weaknesses (e.g., it describes incremental work), and it can significantly benefit from another round of revision. However, I won't object to accepting it if my co-reviewers champion it.

**Missing References:**

Ren and Xiong, ACL 2021: Methods for dealing with both cognitive indicators (EEG) and text. The accuracy of this paper should also be included in the comparison.


**Paper Topic And Main Contributions:**

In this paper, the authors present an approach by aligning EEG and text information to create new features, subsequently utilizing these aligned features to carry out downstream tasks.
The authors conclude that the accuracy of downstream tasks has been enhanced in comparison to other methods that do not employ alignment, and provide some interpretations regarding the aligned EEG features.

**Questions For The Authors:**

1. It was confusing that words that can be used negatively or positively are included in the negative or positive: e.g. an "about" in Negative and an "a" in Positive. What does this mean and what is the interpretation? I found it very difficult to understand how Figures 2 to 4 were made and what the authors were trying to say (partly because of no statistical analysis as I described above).
2. How the authors preprocessed EEG features? It is surprising to obtain near perfect accuracy using EEG (even exceeds that of text information). Could factors other than brain activity be contributing? Is there any possibility that potential artifacts contributed? I would like to see a discussion on how the artefacts are being removed.


**Reasons To Accept:**

1. A multi-modal alignment between brain features and text description has been rightly emphasized for its practical relevance.
2. The paper demonstrates that the accuracy in downstream tasks exceeds other non-alignment methods.
3. The authors have provided the code for analyses.


**Reasons To Reject:**

The interpretation section may benefit from further clarity and depth.
1. While qualitative interpretations are present in Figs 2, 3, and 4, it is hard to understand the implications of these figures. It is nice to include statistical analysis (currently nothing) to support the authors' argument.
2. Furthermore, although the authors provided (only qualitative) interpretations about brain function from the obtained EEG topography, it is well known that such an locational interpretations are inherently difficult in EEG. This is a limitation of this study.


**Reproducibility:**

5: Could easily reproduce the results.

**Reviewer Confidence:**

3: Pretty sure, but there's a chance I missed something. Although I have a good feel for this area in general, I did not carefully check the paper's details, e.g., the math, experimental design, or novelty.

**Typos Grammar Style And Presentation Improvements:**

The attainment of near perfect accuracy using EEG (even exceeds that of text information) is, to be honest, super surprising given that brain signal from EEG is known as very noisy measurement. A discussion on the author's rationale behind this remarkable result would be helpful.

---

> ### Author Rebuttal · Authors · 2023-08-28
>
> Thank you for your valuable feedback! We appreciate it for recognizing our work “A multi-modal alignment between brain features and text description has been rightly emphasized for its practical relevance”. Please find our detailed response to your concerns outlined below.
>
> 1. ***Statistical analysis of Figs 2, 3, 4, and more clarifications***
>
> - Thank you for the suggestion.
> - Figures 2 and 3 illustrate the alignment between EEG signals and individual words at the word level. Given that the datasets solely offer sentence-level annotations and lack word-level annotations, we proceeded by presuming that entire sentences are labeled as "Positive/Negative." However, we acknowledge that certain words likely contribute more informative value than others within these sentences. For instance, words like "romantic" are likely more relevant for EEG alignment than generic words like "be." Regrettably, due to the absence of granularity in the dataset annotations, we were compelled to assume uniform labeling ("Positive/Negative") for all words within sentences. It's important to note that this approach may not capture nuances accurately and could be considered a limitation originating from the dataset's characteristics.
> -To enhance the clarity of our explanations, we have introduced an additional statistical analysis focused on the alignment of words labeled as "Positive/Negative" at the word level. We present the outcome of this analysis in the following Table A.
>
> Table A. Statistical analysis of EEG-word alignment.
> |Method | theta1 | theta2 | alpha1 | alpha2 | beta1  | beta2 |gamma1 |gamma2|
> |:--------: |:--------:| :--------:|:--------:|:--------:|:--------:|:--------:|:--------:| :--------:|
> |Negative| -0.16| -0.32| -0.15| -0.09| -0.18| 0.45| 0.62| 0.67|
> |Positive| -0.18| -0.35| -0.13| -0.08| -0.29| 0.08| 0.39| 0.43|
> |Difference (Negative - Positive)| 0.02| 0.03| -0.02| -0.01| 0.11| 0.37| 0.23| 0.22|
>
> - Table A presents the averaged results of EEG-word alignment across 8 frequency bands, obtained by averaging over 100 EEG-sentence pairs. The analysis highlights noteworthy differences in the alignment outcomes for the "Positive" and "Negative" categories, particularly within the beta2, gamma1, and gamma2 frequency bands. These findings suggest that these specific frequency bands might hold significance, as they exhibit more pronounced differences between the positive and negative sentiments. This observation could potentially stem from the anticipated rise in theta band power during heightened language processing activities, coupled with the theta band's known association with the retrieval of semantic memories.
>
> 2. ***EEG preprocessing and EEG results***
>
> - Regarding the K-Emocon dataset, it's worth noting that the authors of the original work did not disclose the precise preprocessing techniques applied to the EEG waves due to "privacy concerns extending beyond the scope of shared information" [1]. However, we are aware that the EEG signals were captured using the Neurosky device. This device inherently conducts internal preprocessing on raw EEG signals to derive attention and meditation levels, along with brainwave band information. Furthermore, the authors incorporated a Fast Fourier Transform analysis to the EEG signals. In our approach, we followed a similar path. We began by normalizing the amplitudes of the EEG signals using the Z-score technique for each frequency. This normalization process was adopted from previous research [2, 3, 4].
> - In the case of the ZuCo Dataset, the original authors performed a comprehensive preprocessing of the EEG signals utilizing various tools and techniques. Notably, they employed Automagic [5] for initial preprocessing steps. To ensure data quality, they detected and removed problematic electrodes that exhibited noise, flatline, or low-frequency signals. The clean_rawdata function from the EEGLab library was used for this purpose. Further processing included high-pass filtering the EEG signals at 0.5 Hz and applying a notch filter in the range of 49-51 Hz to eliminate power line interference. Eye-related artifacts were addressed by regression of the EOG channels. The authors also integrated the MARA tool to automate the rejection of artifacts, and any electrodes presenting issues were interpolated. A critical alignment step was carried out by synchronizing the EEG data with eye-tracking information using the "EYE EEG extension," a process crucial for precise temporal alignment. For a more in-depth analysis, the EEG signals underwent frequency band filtering, followed by a Hilbert transform to derive power dynamics. A notable analysis point was the comparison of power between frontal left and right electrodes. Stringent criteria were established to identify and exclude artifacts exceeding ±90μV. Following these steps, akin to the K-EmoCon dataset and prior research [2, 3, 4], we proceeded to normalize the preprocessed EEG signals using the Z-score method.
> - The unexpectedly high performance achieved by the EEG signals in our results has sparked our curiosity as well. We postulated that this success might be attributed to the high quality of the collected EEG signals present in the datasets. Additionally, the relatively modest size of the EEG datasets could potentially contribute to the favorable outcomes we observed. This combination of high-quality data and a manageable dataset size likely plays a role in enabling our transformer-based architecture to excel in performance. We are of the opinion that the notable results observed on the Zuco dataset can primarily be attributed to the meticulous collection and processing of the EEG data. In comparison to the baseline established by Hollenstein et al. (2021), where a single RNN/CNN model yielded scores surpassing 70%, the introduction of more powerful transformer architectures naturally holds the potential for even more impressive outcomes. It's worth noting that the K-EmoCon dataset might entail more inherent noise when compared to the Zuco dataset, which could contribute to the slightly lower achieved score of around 70%. This suggests the existence of opportunities for further enhancement in our results.
>
> > [1] Park, Cheul Young, et al. “K-EmoCon, a Multimodal Sensor Dataset for Continuous Emotion Recognition in Naturalistic Conversations.” Scientific Data, vol. 7, no. 1, 8 Sept. 2020, https://doi.org/10.1038/s41597-020-00630-y.
> > [2] Du, Yang, et al. “EEG Temporal–Spatial Transformer for Person Identification.” Scientific Reports, vol. 12, no. 1, 23 Aug. 2022, https://doi.org/10.1038/s41598-022-18502-3.
> > [3] Fdez, Javier, et al. “Cross-Subject EEG-Based Emotion Recognition through Neural Networks with Stratified Normalization.” Frontiers in Neuroscience, vol. 15, 3 Feb. 2021, https://doi.org/10.3389/fnins.2021.626277. Accessed 27 Aug. 2021.
> > [4] S. Yousif, Enas, et al. “Electroencephalogram Signals Classification Based on Feature Normalization.” IOP Conference Series: Materials Science and Engineering, vol. 928, no. 3, 1 Nov. 2020, p. 032028, https://doi.org/10.1088/1757-899x/928/3/032028.
> > [5] Automagic [https://github.com/methlabUZH/automagic](https://github.com/methlabUZH/automagic)
>
> 3. ***Missing References***
>
> Thank you for pointing it out! We will add it in our revision!

---

### Official Review · Reviewer_9CNo · 2023-08-09

**Soundness:** 2

**Excitement:**

2: Mediocre: This paper makes marginal contributions (vs non-contemporaneous work), so I would rather not see it in the conference.

**Paper Topic And Main Contributions:**

This paper proposes a model for predicting stimulus features (e.g, emotion categories and relations between entities) from EEG brain responses. The model embeds stimulus words with a pre-trained language model (BERT-base), uses a series of transformer blocks to encode the stimulus word embeddings and EEG recordings, and then predicts stimulus features from the outputs of the transformer blocks. The authors compare to a set of other models, and show that their models performs better on the task of predicting stimulus features. The model includes a module that computes an alignment between EEG and stimulus words, and the authors analyze this module to determine the types of words that are most "aligned" to different sets of EEG frequency bands. The authors also perform analyses to determine which EEG frequency bands contribute most to prediction performance, and which brain areas are activated in response to more positive or more negative words.

**Questions For The Authors:**

A. How are multimodal features incorporated into MTAM?

B. Do you think this architecture is particularly well-suited for EEG? What are the benefits of specifically analyzing EEG data, rather than using datasets in other neuroimaging modalities (with perhaps a larger number of baselines)?

**Reasons To Accept:**

The paper includes a comparison to multiple baseline models. The results show better predictions for the authors' model across different task conditions.

**Reasons To Reject:**

Clarity of contributions:
In general, it is unclear what the presented results tell us about language processing in the brain, or about language representations in artificial models. For instance, it is unclear what the significance of the MTAM architecture is. What is the purpose of comparing different alignment mechanisms, and of using this particular model architecture? Without further analyses/discussion, the contribution of the model itself is unclear. Although the improvement in performance holds over a number of task conditions, without further analyses/discussion SoTA performance alone does not clarify the purpose of this model architecture. Moreover, given that most of the baselines were not developed for this task, it is unclear whether the improvement is itself a substantial improvement.

Clarity of method:
The model that the authors propose is relatively straightforward, but the explanation of the model (particularly in the main paper) is unclear. For instance, in Table 1 the authors reference three versions of their model ("Ours-EEG", "Ours-Text", and "Ours-Multimodal"), but I was not able to find an explanation of the differences in the inputs for each of these models. Does "Ours-Multimodal" incorporate multimodal stimulus features in the Text Encoder of MTAM, or does "Our-Multimodal" refer to a different type of model?

Soudness and novelty of conclusions:
The authors make some claims as to the brain areas and EEG frequency bands that are aligned/activated in response to positive and negative words. However, the descriptions of the analysis method are too vague to determine whether the conclusions are sound. For instance, it is unclear how positive and negative connotations were labeled, how the set of active frequency bands were determined for each word category (e.g., what level of increase in activity was deemed a reliable difference?), and how brain areas were segmented.

**Reproducibility:**

4: Could mostly reproduce the results, but there may be some variation because of sample variance or minor variations in their interpretation of the protocol or method.

**Reviewer Confidence:**

4: Quite sure. I tried to check the important points carefully. It's unlikely, though conceivable, that I missed something that should affect my ratings.

**Typos Grammar Style And Presentation Improvements:**

In general, the methods are underspecified (please see "Reasons to Reject" for more details).

Some of the terminology is unclear. For instance, the term "connectivity" is used many times. Given other popular uses of the word "connectivity" in the context of neuroimaging data (e.g., "functional connectivity" or "model connectivity"), this term is a bit misleading, especially without a definition.

---

> ### Author Rebuttal · Authors · 2023-08-28
>
> Thank you for the valuable feedback and encouraging review! Please find our detailed response to your concerns outlined below.
>
> 1. ***Clarity of contributions***
>
> We would like to emphasize our primary contributions and provide some clarifications:
>
> > ***Our contributions.***
>
> - Our main goal is to use computational models to discover the connection between language and brain signals, which proves the existence and shows meaningful findings related to cognition.
>
> > ***What is NOT our main contribution***
>
> - Developing the most advanced or elaborate model to attain state-of-the-art (SOTA) outcomes through EEG and text-based classification.
> - While our current model does outperform baselines, our primary emphasis is not centered around achieving optimal classification results. Instead, our intention is to employ these outcomes to illustrate the genuine presence of a connection between EEG patterns and human language. We aim to demonstrate that computational models offer a valuable avenue for exploring this connection.
>
>
> 2. ***"Ours-EEG", "Ours-Text", and "Ours-Multimodal"***
>
> We appreciate your inquiry. Just like demonstrated in Figure 2 of the paper, our proposed approach encompasses two distinct encoders: one dedicated to EEG processing, and another for text processing. In this context, the terms "Ours-EEG" and "Ours-Text" refer to instances where the model exclusively receives EEG or text inputs respectively. Conversely, "Ours-Multimodal" indicates that the model processes both inputs simultaneously. By examining "Ours-EEG" and "Ours-Text," we conduct an ablation study to assess the impact of utilizing only one modality input. This comparison helps unveil insights into our findings. We will add more clarifications in the Appendix in our revision.
>
> 3. ***Soudness and novelty of conclusions***
>
> > How positive and negative aonnotations were labeled?
>
> The annotations indicating positive and negative sentiments were provided by the authors of the datasets [1-2] that we employed in our research. It's important to note that we didn't undertake the task of labeling the data personally.
>
> [1] Park, C.Y., Cha, N., Kang, S., Kim, A., Khandoker, A.H., Hadjileontiadis, L.J., Oh, A.H., Jeong, Y., & Lee, U. (2020). K-EmoCon, a multimodal sensor dataset for continuous emotion recognition in naturalistic conversations. Scientific Data, 7.
> [2] Hollenstein, N., Troendle, M., Zhang, C., & Langer, N. (2019). ZuCo 2.0: A Dataset of Physiological Recordings During Natural Reading and Annotation. International Conference on Language Resources and Evaluation.
>
> > How the set of active frequency bands were determined for each word category (e.g., what level of increase in activity was deemed a reliable difference?)
>
> As detailed in Section 4.3, we meticulously examined all the provided frequency bands of the dataset, as evidenced in Appendix Section D.3. Our objective was to visualize the brain regions exhibiting the most significant activity, particularly within the regions of interest – Wernicke's and Broca's areas. These areas were specifically chosen due to their prominent involvement in language processing. After scrutinizing the complete brain topological maps, it became evident that beta2, gamma1, and gamma2 frequencies exhibited the highest amplitude in microvolts within these regions. Consequently, we opted to showcase these three frequencies as representative maps in our paper.
>
> > How brain areas were segmented
>
> In terms of the segmentation of the brain areas, we closely followed the description of credible sources [3-5] for the locations of the areas of interest (Broca's area, Wernicke's area, occipital lobes, inferior parietal lobes).
>
> [3] University of California San Francisco. “Speech & Language.” Memory and Aging Center, 2019, memory.ucsf.edu/symptoms/speech-language.
> [4] “Inferior Parietal Lobule - an Overview | ScienceDirect Topics.” Sciencedirect.com, 2014, www.sciencedirect.com/topics/neuroscience/inferior-parietal-lobule.
> [5] “Brain Map: Occipital Lobes | Queensland Health.” Www.health.qld.gov.au, 22 Jan. 2021, www.health.qld.gov.au/abios/asp/boccipital.
> ‌
>
>
> 4. ***How are multimodal features incorporated into MTAM?***
>
> Please refer to Figure 1 in our paper. Our model intricately investigates EEG and language features through the utilization of two encoders. On the left, we have the EEG encoder, while on the right, the language encoder is depicted. To establish connections and relationships between the two domains, we introduce the cross-alignment module. This module effectively learns the connectivity and associations between EEG and language, resulting in the generation of multimodal features.
>
> 5. ***Do you think this architecture is particularly well-suited for EEG? What are the benefits of specifically analyzing EEG data, rather than using datasets in other neuroimaging modalities (with perhaps a larger number of baselines)?***
>
> - Thank you for the question. Our choice to focus on EEG signals stems from its simplicity as a means of uncovering brain activity. Collecting signals from sources like brain-MRI can be considerably more expensive, posing potential challenges for subsequent research. Furthermore, EEG signals hold a paramount position in cognitive and computational neuroscience studies, being the most extensively utilized brain signal. This very reason underpins our decision to employ EEG as the material in our work, representing the first effort to delve into the intricate interplay between brain signals and human language via a computational model.
> - Another crucial aspect we wish to emphasize is the availability of a substantially larger number of existing EEG datasets in comparison to neuroimaging datasets. Within our specific context, it's essential to collect brain signals in tandem with human language during the course of human subject experiments. This simultaneous collection of brain signals and human language is conducted within highly granular timestamps to ensure a detailed alignment between the two.
> - Under the stringent criterion above, a significant proportion of the available EEG datasets fall short of meeting these specific requirements. Moreover, the situation is even more constrained for neuroimaging datasets, which primarily encompass brain signals without the incorporation of human languages. Even within the EEG modality, our research has led us to identify merely two publicly accessible datasets that conform to these stringent prerequisites – these are precisely the two datasets [6,7] that we've utilized in our study.
> - We appreciate the reviewer for the great question. Taking feasibility into careful consideration, EEG emerges as the optimal choice for conducting this study.
>
> > [6] Park, C.Y., Cha, N., Kang, S., Kim, A., Khandoker, A.H., Hadjileontiadis, L.J., Oh, A.H., Jeong, Y., & Lee, U. (2020). K-EmoCon, a multimodal sensor dataset for continuous emotion recognition in naturalistic conversations. Scientific Data, 7.
> > [7] Hollenstein, N., Troendle, M., Zhang, C., & Langer, N. (2019). ZuCo 2.0: A Dataset of Physiological Recordings During Natural Reading and Annotation. International Conference on Language Resources and Evaluation.
>
> 6. ***Terminology: "connectivity"***
>
> Thank you for the question. In our context, the term "connectivity" pertains to the interrelation and interdependence between EEG signals and language processing (L006-L007). Given the innovative nature of our research, which delves into the uncharted territory of exploring the link between EEG and human language, it's possible that an exact term from existing literature might not precisely encapsulate our setting.
>
> 7. ***Reproducibility***
>
> We would respectfully inquire whether the reviewer identifies any aspects they believe are insufficient for reproducing our work. The code and data necessary for replication can be accessed through an anonymous link: [https://anonymous.4open.science/r/EMNLP_2023-B08D/](https://anonymous.4open.science/r/EMNLP_2023-B08D/) (mentioned in paper lines L033-L034). This link encompasses the complete setup, including environment configuration, raw and processed data, along with scripts for processing, training, testing, and visualization. We are confident that these resources facilitate result reproduction. If there are any additional materials you require, please do not hesitate to inform us.

---

### Official Review · Reviewer_fE1A · 2023-08-11

**Typos Grammar Style And Presentation Improvements:** NA as far as I know.
**Soundness:** 3

**Excitement:**

4: Strong: This paper deepens the understanding of some phenomenon or lowers the barriers to an existing research direction.

**Missing References:**

NA as far as I know.

**Paper Topic And Main Contributions:**

Authors propose a transformer-based multi-modal model that exploits language and EEG information to perform downstream tasks, such as sentiment analysis and relation detection. The study claims to be the first, after a prior work that is relatively related, cited as y Hollenstein et al. (2021)’. This is as baseline, and some other works prior to 2016 are also used to compare the method with them.

**Questions For The Authors:**

Have mentioned earlier.

**Reasons To Accept:**

The paper is of course one of the firsts, to relate EEG to language. Codes are available and as far as I checked they look clear and enough for repro. So the merit of being the first in the area seems to be useful in the first sight.

**Reasons To Reject:**

How you argue table 1, in the sense that values reported even with only EEG are very high, and together with text they become close to 100 % accuracy on SA and SD tasks?
There are speculations about relationship of the brain signals such as EEG to computerized information as in the tokens of the language. There is a little information on how these signals can expose information about what the person is thinking or talking.
I would expect to see some discussions on this aspect at least to show some human studies or refer for interested readers.

**Reproducibility:**

5: Could easily reproduce the results.

**Reviewer Confidence:**

3: Pretty sure, but there's a chance I missed something. Although I have a good feel for this area in general, I did not carefully check the paper's details, e.g., the math, experimental design, or novelty.

---

> ### Author Rebuttal · Authors · 2023-08-28
>
> Thank you for your valuable comments! We appreciate it for recognizing our work “The paper is of course one of the firsts, to relate EEG to language. The merit of being the first in the area seems to be useful in the first sight”. Please find our detailed response to your concerns outlined below.
>
> 1. ***Relationship of the brain signals such as EEG to human language***
>
> - Thank you for the valuable question!
> - Concerning the enhanced outcomes from the fusion of EEG and text data compared to the results of each modality individually, this is a key observation outlined in our work. Furthermore, it's noteworthy that the reported outcomes solely based on EEG data demonstrate remarkable performance, potentially indicating the quality of EEG signals within this dataset is very good. These signals could already offer utility in subsequent classification tasks. However, it's important to highlight that our primary objective doesn't revolve around introducing a model to achieve state-of-the-art results. Instead, our central goal is to uncover the inherent relationship between EEG patterns and human language, showcasing the potential for exploration using computational models.
> - Regarding our analysis, we have observed a distinct functional division where Broca's area is implicated in speech production, while Wernicke's area is linked to speech comprehension. These conclusions are based on the outcomes of our experimental investigations. It's worth acknowledging, however, that the field of research surrounding this matter might not have reached a definitive consensus. In fact, multiple researchers have put forth contrasting hypotheses. Pulvermüller (2018), Strijkers & Costa (2016), Pylkkänen (2019), and Friederici (2011) [1-4] have each proposed brain-language models that suggest both frontal regions, like Broca's area, and temporal regions, like Wernicke's area, play active roles in both speech production and perception. Conversely, Matchin and Hickok (2020) [5] propose an alternative viewpoint, suggesting that Broca's area is exclusively involved in production and not perception, whereas temporal regions are posited to contribute to both production and perception.
> -  As discussed in [6], Kensinger outlined the general tendency for negative events to be more memorable compared to positive events. Expanding upon the concepts presented in [1], it is suggested that negative words possess the capability to create stronger and enduring memories than positive words. Consequently, these negative words might exhibit heightened activation within the occipital and inferior parietal lobes.
> - In our examination, we identified heightened activation in both the occipital lobes and slightly extending over the inferior parietal lobes in response to negative words, similar to the patterns observed for positive words. These specific brain regions have been previously associated with functions such as visual processing, visuospatial attention, and semantic memory.
> - The intricate connection between brain signals like EEG and human language is undeniably complex. To address this complexity, we introduced a computational approach with the intention of delving into this relationship and offering a novel avenue for its exploration.
>
> >[1] Friederici, A. D. (2011). The brain basis of language processing: From structure to function. Physiological Reviews, 91(4), 1357–1392.
> Matchin, W., & Hickok, G. (2020). The Cortical Organization of Syntax. Cerebral Cortex, 30(3), 1481–1498.
> > [2] Pulvermüller, F. (2018). Neural reuse of action perception circuits for language, concepts and communication. Progress in Neurobiology, 160, 1–44.
> > [3] Pylkkänen, L. (2019). The neural basis of combinatory syntax and semantics. Science, 366(6461), 62–66.
> > [4] Strijkers, K., & Costa, A. (2016). The cortical dynamics of speaking: present shortcomings and future avenues. Language, Cognition and Neuroscience, 31(4), 484–503.
> > [5] Matchin, W., & Hickok, G. (2019). The Cortical Organization of Syntax. Cerebral cortex.
> > [6] Kensinger, Elizabeth A.. “Remembering the Details: Effects of Emotion.’’ Emotion Review 1 (2009): 113 - 99.

---

### Meta-Review · Area_Chair_9dVH · 2023-09-04

**Recommendation:** 1
**Confidence:** 5

**Metareview:**

The paper in submission explores the connection between EEG (Electroencephalogram) signals and language, making it one of the first studies to do so in the field. The authors provide code for analyses, making the research reproducible. The study uses multiple baseline models for comparison and shows that their model outperforms others across different task conditions. A significant strength lies in emphasizing the multi-modal alignment between brain features and text descriptions, highlighting its practical relevance. The paper demonstrates superior accuracy in downstream tasks compared to non-alignment methods.

However, the paper also has several shortcomings that raise questions about its overall contribution and clarity. One major concern is the near-100% accuracy reported for certain tasks, which seems implausible and is not adequately explained. There is also a lack of information regarding how EEG signals can be reliably interpreted to understand what a person is thinking or talking about. The study lacks a clear articulation of its contributions to our understanding of language processing in the brain or language representations in artificial models.

The methodological presentation in the paper is somewhat weak. It references three versions of their model ("Ours-EEG," "Ours-Text," and "Ours-Multimodal") without clearly explaining the differences in their inputs. Questions about the model architecture remain unanswered, reducing the clarity of the paper's contributions. Concerns are also raised about the soundness and novelty of the conclusions, particularly relating to the identification of brain areas and EEG frequency bands responding to positive and negative words. The description of the methodology used is considered too vague to affirm whether the conclusions are valid.

The paper could benefit from additional statistical analyses to support its arguments, as only qualitative interpretations are currently provided. The limitations of using EEG for locational interpretations are acknowledged but not thoroughly discussed. The paper also fails to answer some high-level questions critical to the field, such as how the continuous EEG data align with the discrete nature of language and how individual subjectivity in EEG signals is addressed.

Further, given that the paper is intended for an NLP (Natural Language Processing) conference, the authors should include explanations of physiological signals like EEG and MEG (Magnetoencephalography) for readers unfamiliar with these concepts. The paper is also advised to delve into why specific metrics like CCA (Canonical Correlation Analysis) and WD (Wasserstein Distance) were chosen over others like KL divergence or Euclidean distance.

Lastly, there are several mistakes in the paper, ranging from undefined acronyms to grammatical errors, that need correction before final submission.

Overall, while the paper makes a pioneering attempt to link EEG with language and offers promising results, it suffers from issues related to clarity, methodology, and the soundness of its conclusions. I would suggest the author submit this work to journals instead of presented in EMNLP 2023.

---

### Decision · Program_Chairs · 2023-10-07

**Decision:**

Accept-Findings

**Comment:**

The paper in submission explores the connection between EEG (Electroencephalogram) signals and language, making it one of the first studies to do so in the field. The authors provide code for analyses, making the research reproducible. The study uses multiple baseline models for comparison and shows that their model outperforms others across different task conditions. A significant strength lies in emphasizing the multi-modal alignment between brain features and text descriptions, highlighting its practical relevance. The paper demonstrates superior accuracy in downstream tasks compared to non-alignment methods.

However, the paper also has several shortcomings that raise questions about its overall contribution and clarity. One major concern is the near-100% accuracy reported for certain tasks, which seems implausible and is not adequately explained. There is also a lack of information regarding how EEG signals can be reliably interpreted to understand what a person is thinking or talking about. The study lacks a clear articulation of its contributions to our understanding of language processing in the brain or language representations in artificial models.

The methodological presentation in the paper is somewhat weak. It references three versions of their model ("Ours-EEG," "Ours-Text," and "Ours-Multimodal") without clearly explaining the differences in their inputs. Questions about the model architecture remain unanswered, reducing the clarity of the paper's contributions. Concerns are also raised about the soundness and novelty of the conclusions, particularly relating to the identification of brain areas and EEG frequency bands responding to positive and negative words. The description of the methodology used is considered too vague to affirm whether the conclusions are valid.

The paper could benefit from additional statistical analyses to support its arguments, as only qualitative interpretations are currently provided. The limitations of using EEG for locational interpretations are acknowledged but not thoroughly discussed. The paper also fails to answer some high-level questions critical to the field, such as how the continuous EEG data align with the discrete nature of language and how individual subjectivity in EEG signals is addressed.

Further, given that the paper is intended for an NLP (Natural Language Processing) conference, the authors should include explanations of physiological signals like EEG and MEG (Magnetoencephalography) for readers unfamiliar with these concepts. The paper is also advised to delve into why specific metrics like CCA (Canonical Correlation Analysis) and WD (Wasserstein Distance) were chosen over others like KL divergence or Euclidean distance.

Lastly, there are several mistakes in the paper, ranging from undefined acronyms to grammatical errors, that need correction before final submission.

Overall, while the paper makes a pioneering attempt to link EEG with language and offers promising results, it suffers from issues related to clarity, methodology, and the soundness of its conclusions. I would suggest the author submit this work to journals instead of presented in EMNLP 2023.